# HIV Infection, Chromosome Instability, and Micronucleus Formation

**DOI:** 10.3390/v15010155

**Published:** 2023-01-04

**Authors:** Joel Henrique Ellwanger, Bruna Kulmann-Leal, Marina Ziliotto, José Artur Bogo Chies

**Affiliations:** Postgraduate Program in Genetics and Molecular Biology (PPGBM), Laboratory of Immunobiology and Immunogenetics, Department of Genetics, Universidade Federal do Rio Grande do Sul (UFRGS), Porto Alegre 91501-970, Brazil

**Keywords:** antiretroviral therapy, chromosomal aberration, DNA damage, immunosenescence, inflammation, chromosome instability, HIV, micronucleus

## Abstract

Genome integrity is critical for proper cell functioning, and chromosome instability can lead to age-related diseases, including cancer and neurodegenerative disorders. Chromosome instability is caused by multiple factors, including replication stress, chromosome missegregation, exposure to pollutants, and viral infections. Although many studies have investigated the effects of environmental or lifestyle genotoxins on chromosomal integrity, information on the effects of viral infections on micronucleus formation and other chromosomal aberrations is still limited. Currently, HIV infection is considered a chronic disease treatable by antiretroviral therapy (ART). However, HIV-infected individuals still face important health problems, such as chronic inflammation and age-related diseases. In this context, this article reviews studies that have evaluated genomic instability using micronucleus assays in the context of HIV infection. In brief, HIV can induce chromosome instability directly through the interaction of HIV proteins with host DNA and indirectly through chronic inflammation or as a result of ART use. Connections between HIV infection, immunosenescence and age-related disease are discussed in this article. The monitoring of HIV-infected individuals should consider the increased risk of chromosome instability, and lifestyle interventions, such as reduced exposure to genotoxins and an antioxidant-rich diet, should be considered. Therapies to reduce chronic inflammation in HIV infection are needed.

## 1. Introduction

The maintenance of genome integrity is a key factor in proper cell functioning and disease prevention. An accumulation of DNA damage accelerates aging by impairing cell metabolism, causing senescence and immunosenescence, apoptosis, stem-cell exhaustion, and inflammation, among other deleterious effects to cells, thus increasing the risk of age-related diseases [1,2]. Genomic instability, which is the tendency of the genome to undergo mutation or chemical modification [1], is caused by replication stress, chromosome missegregation via defective mitosis, impaired homologous recombination, environmental insults (e.g., exposure to pollutants, pesticides, or radiation), and lifestyle factors (e.g., diet, physical activity, smoking, and alcohol consumption). Genomic instability can result in a loss or amplification of genes, rearrangements, extrachromosomal DNA, and micronuclei formation, among other molecular breakdowns, with multiple pathological consequences, including various types of cancer [3,4,5].

The micronucleus (MN) is a small and rounded DNA-containing nuclear structure observed to be isolated in the cytoplasm, located adjacent to the main nucleus. Mitotic errors or DNA damage can produce lagging chromosomes or chromosome fragments, which are deposited in the cytoplasm in the form of a MN [6]. Increased MN frequency is associated with age, occupational exposure to genotoxins, a heavy smoking habit, and cancer [7,8]. Micronuclei are well-accepted biomarkers of DNA damage and chromosomal instability in multiple organisms, including humans [9].

The rupture of the nuclear envelope of a MN and the consequent exposure of DNA to the cytoplasm has deleterious consequences for cells, such as inflammation and chromothripsis [6]. Of note, chromothripsis is a mutational process that promotes cancer development due to tumor suppressor loss, oncogenic translocations, or oncogene amplification [10]. Inflammation derived from MN formation and rupture is an increasingly recognized issue, associated with various diseases. Therefore, micronuclei are biomarkers of DNA damage and chromosomal instability, as well as inducers of hypermutation and inflammation [9].

Viruses can also damage the genetic material and impair DNA repair processes in host cells directly by interaction with DNA and the proteins of DNA repair machinery, and indirectly by the production of reactive oxygen species (ROS) and oncoproteins, replication stress, exacerbation of inflammatory responses, and impairment of cellular integrity or function. Viruses are therefore recognized inducers of genome instability in host cells, with some viral species (Epstein–Barr virus, hepatitis B virus, Kaposi’s sarcoma-associated herpesvirus, and human papillomaviruses, among others) inducing carcinogenesis in humans [11,12].

Retroviruses damage DNA through multiple mechanisms, including genome integration, replication, inflammation, and the direct interaction of viral proteins with DNA. For example, HIV-1 lentiviral protein Vpr induces single-strand DNA breaks (SSBs) and double-strand DNA breaks (DBSs), potentially by inducing replication fork collapse after the inhibition of DNA replication [13]. In addition to promoting SSBs and DBSs, Vpr and other HIV-1 proteins (Tat and Vif) may damage DNA by other mechanisms, including by repressing DNA damage response (DDR) and DNA repair [13]. As a consequence, HIV Vpr can induce MN formation and other chromosomal aberrations [14,15]. Of note, HIV proteins Tat, Vif, and Vpr promote cellular arrest, potentially benefiting viral replication, and latent HIV-infected cells are more susceptible to DNA damage [13]. Figure 1 summarizes the connections between HIV proteins and MN formation.

HIV drugs are also associated with chromosome instability [16]. People living with HIV, even on antiretroviral therapy (ART), suffer from chronic diseases and conditions associated with aging (e.g., cancer and cardiovascular and neurodegenerative diseases) at an increased rate compared to uninfected people. This indicates that HIV facilitates the occurrence of age-related diseases, independently or in association with other risk factors, potentially through the exacerbation of chronic inflammation, cellular senescence, mitochondrial dysfunction, telomere attrition, stem-cell exhaustion, DNA damage, and genomic instability [17]. Chronic inflammation is a major issue for HIV-infected individuals, and is caused by dysbiosis, microbial translocation, co-infections, and residual HIV replication, among other factors. As mentioned above, these conditions are commonly observed even in HIV-infected individuals undergoing ART [18,19,20,21].

Micronuclei formation and the release of DNA into the cytosol can trigger inflammatory cascades [22]. Infection-related inflammation produces reactive oxygen species (ROS) and reactive nitrogen species which, on one hand, are important in protecting the host from pathogens but, on the other hand, cause DNA damage. Consequently, DNA damage and DDR trigger further inflammation, creating a vicious circle of inflammation and genetic damage [22]. Other cellular processes also influence MN formation during HIV infection. Autophagy promotes cellular recycling [23,24] and can limit the amount of cytoplasmic DNA derived from DNA insults (e.g., MN can be degraded by the autophagy–lysosomal pathway), thus determining cell fate following genomic instability events. In brief, autophagy is a protective mechanism against MN formation and related consequences [25,26] and this can explain, at least partially, the “genome-stabilizing effects of autophagy” [27]. Autophagy also contributes to the maintenance of the innate immune homeostasis, protecting humans from inflammatory conditions [24]. HIV can inhibit autophagy [23,28,29,30] and, consequently, such inhibition contributes to chromosome instability and chronic inflammation. Deciphering the connections between viral infection, genome instability, and DDR can help understandings of viral pathogenesis and the development of better antiviral therapies [13].

The buccal micronucleus cytome (BMCyt) assay is a method used to study chromosomal instability, DNA damage, and cell death using cells exfoliated from oral mucosal tissue, based on the microscopic analysis of cytoplasmic and nuclear morphology [31]. The BMCyt assay is considered a minimally invasive method [31]; therefore, it is widely used by laboratories from different parts of the world to evaluate genetic damage in human populations [8]. The BMCyt assay yields the quantification of the frequency of chromosome breakage or loss due to incorrect mitosis (observed microscopically in the cytoplasm as a MN), gene amplification (viewed as nuclear buds), cytokinesis failure or arrest (observed as binucleated cells), and cell death (observed as condensed chromatin, karyorrhectic, pyknotic, or karyolitic cells) [8,31]. Of note, MN frequency in exfoliated buccal cells highly correlates with MN frequency in peripheral blood lymphocytes [8]. The estimated spontaneous MN frequency in (healthy) human buccal exfoliated cells is 0.74‰ (95% CI 0.52–1.05) [8]. The BMCyt assay is commonly called the “MN test” or “MN assay”, and variations of this technique also exist, including the evaluation of micronuclei and other nuclear anomalies in non-buccal epithelial cells (e.g., nasal and cervical cells) [9].

Considering the information mentioned above and the poorly explored connections between HIV infection, inflammation, and chromosomal instability, we reviewed studies from multiple countries that evaluate genomic instability, using MN as a biomarker, in the context of HIV infection. This narrative review also discusses the connections between HIV infection, immunosenescence, and age-related diseases.

## 2. Impacts of HIV Infection and Treatment on Chromosomal Integrity: A Focus on Human Studies

In Brazil, Lima et al. [32] investigated micronuclei in exfoliated oral cells of HIV-infected individuals undergoing ART and non-infected controls (*n* = 30 each group). In addition to measuring MN frequency, the authors separated micronuclei into two categories: (I) single MN and (II) multiple micronuclei. The total number of micronucleated cells and the MN number were not statistically different between groups, and no statistical correlation between CD4+ T cell counts, and MN frequency was observed. Considering the two MN categories mentioned, a statistically significant increased mean of single MN in the cells of the controls compared to those of HIV-infected individuals and a non-significant increase in the occurrence of multiple micronuclei in the cells of the HIV group compared to the controls were reported. However, the small sample size of this study and the absence of difference in overall MN frequency between the groups may limit interpretations of these findings [32].

In South Africa, Baeyens et al. [33] evaluated the chromosomal radiosensitivity of HIV-infected individuals (*n* = 49) and controls (*n* = 29) using the MN assay. Blood samples from both groups were exposed to doses of 6 MV X-rays in vitro, ranging from 1 to 4 Gy. MN frequencies were significantly higher in irradiated lymphocytes of HIV-infected individuals compared to the controls at all exposure doses (1, 2, 3, and 4 Gy), which can be at least partially attributed to differences in CD4+/CD8+ T cell ratios between HIV-infected individuals and controls, in addition to the direct effects of HIV on chromosomal instability (increased DNA damage and impaired DNA repair and apoptosis, among others). In brief, this result suggests that cells of HIV-infected individuals have increased radiosensitivity [33]. Subsequently, other studies confirmed these findings, also describing increased MN frequencies in the blood cells of South African HIV-infected individuals exposed to radiation in vitro [34,35]. Increased radiosensitivity can be problematic, especially in HIV-infected individuals with cancer who need to undergo radiotherapy [35].

Zizza et al. [36] evaluated MN frequency in peripheral blood mononuclear cells of HIV-infected individuals undergoing ART (*n* = 52) and non-infected controls (*n* = 55), and both groups were from Italy. They found an increased MN frequency in the HIV-infected group, with HCV co-infection and HIV-RNA load being risk factors for increased MN frequency. Individuals with undetectable viremia showed a reduced MN frequency compared to those with uncontrolled viremia. These results indicate that MN in HIV-infected individuals undergoing ART is not a feature exclusively derived from ART or HIV infection per se, but that it is also due to problems associated with chronic infection (i.e., co-infections). However, the absence of a group composed of HIV-infected but ART-naive individuals precludes the inferring of the degree of contribution of HIV infection and ART use on MN frequency, separately [36].

Several in vitro studies, as well as studies performed in animals, have evidenced genotoxic effects (e.g., formation of MN and nucleoplasmic bridges) of various drugs used to treat HIV infection, including zidovudine, tenofovir disoproxil fumarate, lamivudine, and efavirenz. Taken together, these reports indicate that HIV drugs can indeed induce clastogenic and aneugenic effects on chromosomes [16,37,38,39,40,41].

Zidovudine-based ART is commonly used to prevent mother-to-child HIV transmissions. In this context, Witt et al. [42] evaluated chromosomal damage in children exposed to zidovudine (transplacentary and post-partum) in a study performed in the USA. Micronucleated reticulocyte frequencies were evaluated by flow cytometry in children (*n* = 16, all subjects received prophylactic post-partum zidovudine for 6 weeks) and mothers on zidovudine-based ART (*n* = 13) or ART without zidovudine (*n* = 3, a small control group) pre-natal. In women, samples were obtained from venous blood. In children, samples were obtained from both cord blood at birth and subsequently from venous blood. Controls were obtained from HIV-negative cord-blood samples (*n* = 10). A 10-fold increase in micronucleated reticulocyte frequencies was observed in mothers and children pre-natal with zidovudine-based ART compared to the controls [42]. These results confirm, in humans, the genotoxic effects of zidovudine observed in vitro [37] and in animals [38].

In India, Shah et al. [43] evaluated MN frequency in oral cells of HIV-infected women undergoing ART and healthy controls (*n* = 25 each group) using Papanicolaou staining smears. The HIV-infected group showed a significant increase in MN rate (~two-fold) compared to the controls. Papanicolaou stain is considered effective in detecting and evaluating micronuclei [43]. However, it is important to stress that an abnormal increase in MN frequency can be observed in non-DNA-specific stains. For example, Giemsa and aceto-orcein stains can produce higher MN frequencies. Feulgen-Fast Green is considered the most specific and versatile DNA-specific staining method; therefore, it is the standard staining method used in the BMCyt assay [8].

In Mexico, Gutiérrez-Sevilla et al. [44] evaluated genomic instability markers in HIV-infected individuals undergoing ART (*n* = 46, with two sub-groups, each receiving a specific ART combination), HIV-infected but ART-naive individuals (*n* = 13), and HIV-negative controls (*n* = 8). Genomic instability markers were accessed by nuclear abnormality analyses (MN, binucleated cells, nuclear buds, karyorrhexis, karyolysis, and pyknosis) of buccal mucosal samples using the BMCyt assay [31]. No difference in MN frequencies was observed between the groups. However, higher frequencies of binucleated cells and nuclear buds were observed in both HIV-infected ART-naive individuals and HIV-infected individuals undergoing ART compared to the HIV-negative control group. This result suggests that the effect of HIV on genomic instability can occur independently of ART. However, differences in genomic instability caused by variations of ART regimens can occur. Karyorrhexis, binucleated cells, and nuclear buds were found to be increased in the subgroup of HIV-infected individuals receiving reverse transcriptase inhibitors (RTIs) as ART compared to the controls. Such a difference was not observed between the subgroup of HIV-infected individuals receiving protease inhibitors (PIs) and controls. This suggest that ART based on PIs can produce less cytotoxic damage than RTIs. Finally, the authors also found a positive correlation between the nuclear buds and CD4+/CD8+ ratio among the HIV-infected individuals, suggesting a role of CD4+ T cells in genomic instability occurrence [44].

As mentioned previously [16,37,38,39,40,41] and reinforced by the results of Gutiérrez-Sevilla et al. [44], different antiretroviral combinations have varying toxic effects on DNA. Since the United States Food and Drug Administration (FDA) approved Zidovudine in 1987 for the treatment of HIV infection [45], many other antiretrovirals have been developed and HIV therapy has greatly advanced, with important improvements in terms of decreased toxicity and side effects. However, even modern antiretrovirals have some cellular toxicity, damaging DNA and mitochondria to some extent. Maraviroc is an antiretroviral approved for clinical use in 2007. This drug inhibits HIV infection by interfering with virus interaction with the human C-C chemokine receptor type 5 (CCR5), the main HIV-1 co-receptor [46]. Other new CCR5 antagonists for the treatment of HIV infection and other conditions (i.e., cancers) are under investigation, including Vicriviroc and Leronlimab [47,48]. The main function of CCR5 is to regulate the activity of inflammatory cells [46], but it has recently been demonstrated that CCR5 also participates in the control of DNA repair [47]. Consequently, CCR5 antagonists (i.e., Maraviroc, Vicriviroc, and Leronlimab) can lead to genomic instability by impairing DNA repair and through other CCR5-related mechanisms [47,48,49]. However, it is important to emphasize that evidence of the participation of CCR5 in DNA damage/repair comes from cancer studies, in which the participation of other drugs is present [47,48,49]. The clinical significance of this effect of CCR5 antagonists on DNA damage in HIV-infected individuals is still speculative. Beyond CCR5 antagonists, other HIV drugs (e.g., Dolutegravir) can cause mitochondrial ROS production, mtDNA damage, mitochondrial dysfunction, and cell death [50,51,52,53].

Lazarde-Ramos et al. [54] observed, also in Mexico, an increased frequency of MN and other nuclear abnormalities (binucleated and karyorrhectic cells) in HIV-infected individuals (*n* = 22) undergoing ART (ATRIPLA: efavirenz, 600 mg; emtricitabine, 200 mg; and tenofovir disoproxil fumarate, 300 mg) compared to a control group (*n* = 22) using the BMCyt assay. In this same study [54], the administration of ART in combination with aqueous (*n* = 22) or methanolic (*n* = 23) extracts of rosemary (*Rosmarinus officinalis*, a plant species with anti-inflammatory and antioxidant properties [55]), significantly reduced the frequency of MN (in the methanolic extract group) and abnormally condensed chromatin, karyorrhexis, and binucleated cells (in both the methanolic and aqueous extract groups), compared to the use of ART alone. These findings suggest that the prescription of antioxidants or plant extracts with antioxidant properties could be beneficial for HIV-infected individuals undergoing ART [54]. However, more studies on these aspects need to be performed before any clinical recommendation, especially considering the potential adverse effects of herbal medicines and herb–drug interactions. Finally, Table 1 summarizes the impacts of HIV infection and treatment on chromosomal integrity based on human studies cited in this section.

### Potential Lifestyle and Nutritional Interventions to Be Used in Association with ART

In addition to the potential benefits of plant-based extracts highlighted by Lazarde-Ramos et al. [54], some lifestyles and nutritional habits can help to control genome instability, thus benefiting HIV-infected individuals undergoing ART. For example, reduced MN frequency is associated with daily fruit consumption [8]. In this sense, proper intake of micronutrients (e.g., antioxidant vitamins, selenium), regular physical activity, and other behavior interventions (e.g., UV protection, cessation of tobacco smoking, and adequate sleep/rest) can help reduce oxidative stress and HIV-related aging manifestations, including genome instability [1,2,17,56,57]. Specifically concerning micronutrients, it is essential to consider the potential deleterious effects of some nutrients, especially when used in inappropriate doses. For example, selenium in excess can be detrimental (even toxic) to humans [58,59]. High vitamin D levels can trigger inflammation in HIV-infected individuals [60], and vitamin D supplementation in individuals with proper plasma levels of this micronutrient does not provide health benefits [61]. Recommendations for the therapeutic use of micronutrients during HIV infection should be made by a qualified health professional (e.g., physician, nutritionist), and it is essential to pay special attention to the prescribed doses for each micronutrient.

Some studies suggest that a diet rich in anti-inflammatory compounds (food-derived fibers, ω-3, magnesium, flavonoids, and carotenoids, among others), commonly observed in Mediterranean diet patterns and ‘plant-based foods’, may help to control chronic inflammation [62,63], being potentially beneficial for HIV-infected individuals undergoing ART concerning inflammatory, metabolic, and immune markers [64,65,66,67]. However, therapeutic strategies (from nutritional therapy to regular drugs) to control inflammation in chronic HIV infection are still limited, indicating the need for more studies to focus on this issue.

## 3. Immunosenescence, HIV Infection and Chromosome Instability

The concept of immunosenescence includes a set of processes that culminate in a weakening of the immune system following the course of aging, associated with increased morbidity and mortality risks [68,69,70]. Immunosenescence-related processes include changes in the innate and adaptive immune systems, with atrophy or involution of the thymus being the first observed and best characterized process related to immunosenescence [69,70]. Thymus involution leads to a decrease in the production of naive T cells, which results in an activation of further replication of pre-existing memory T cells in an attempt to maintain a meaningful and still diverse repertoire [71,72]. Such increased replication results in ‘replicative senescence’, in which cells lose the ability to replicate over time until they reach exhaustion [69,70]. T cell exhaustion also occurs when cells are chronically exposed to high levels of antigens, leading to severe T cell dysfunction. This exhaustion state triggers a deficient immune control of HIV infection. Consequently, chronic HIV infection creates a vicious circle of infection-associated antigen production and a loss of control of HIV infection [73]. Of note, inhibitory signals of T cell activation (e.g., PD-1, TIGIT, LAG-3) [73] are linked to T cell exhaustion, persistent HIV infection, and disease progression, even in individuals undergoing ART [74,75,76]. Following thymus involution, the secretion of pro-inflammatory cytokines occurs, which correlates with a state of chronic inflammation and, at the same time, a greater susceptibility to infections, autoimmunity, and cardiovascular diseases, as well as other outcomes present in elderly individuals. Furthermore, HIV-infected individuals with immunosenescence features show an acceleration of progression to AIDS [69,70].

The profile of immune cells can be used as indicators of aging and to determine the immunosenescence state [77]. The epigenetic clock (i.e., DNA methylation data, DNA methylation-based estimate of telomere length) is a useful biomarker in detecting HIV-related aging, a marker that usually indicates accelerated aging in HIV-infected individuals [78,79]. Furthermore, telomere length is a pivotal marker of replicative history and an indicator of biological aging, and telomere shortening is associated with chromosome instability and immunosenescence [80,81,82]. In this context, multiple studies have shown that inflammation, immune activation, and other factors related to HIV infection are associated with telomere shortening, which can contribute to immunosenescence, aging, and age-related diseases in HIV-infected individuals [80,83,84,85,86,87,88]. The ‘oxi-inflamm-aging theory’ is a concept of aging that describes this biological process as an association of chronic, low-grade inflammation in association with oxidative stress, which prejudices the homeostasis of the nervous, endocrine, and immune systems. This culminates in higher morbidity and mortality [89].

Many causes of ‘inflamm-aging’ are related to chromosome instability, such as defective autophagy/mitophagy, the activation of inflammasome by cell debris and misplaced self-molecules, and DDR activation [90]. Cell senescence is, at least partially, a result of chromosomal damage accumulation and defective cell-cycle functioning. Genome instability, expressed by nuclear anomalies and chromosomal aberrations, can lead to defective cell division, apoptosis, cell-cycle arrest, and carcinogenesis [91,92,93]. Furthermore, MN frequency increases with aging and is associated with less proliferative lymphocytes [94]. Additionally, increased genomic instability and MN formation have an impact on lymphocyte function and immunosenescence [93]. Chromosome instability and immunosenescence are therefore connected processes that can form a vicious circle, which can be aggravated by HIV infection.

Immunoscenence is also associated with insertions of mitochondrial sequences into the nuclear DNA of T lymphocytes, which increased progressively with aging [93]. ROS generation in the mitochondria (mitROS) could damage/fragment mtDNA, with such fragments accumulating into the nuclear genome. mtDNA insertions may compromise chromosome segregation and cause increasing genomic instability and MN formation in lymphocytes, forming cells with a reduced proliferation capacity and an increase in apoptosis, which are typical immunosenescence markers. These processes may affect the balance of cell division, differentiation, senescence, and death, which is essential for the maintenance of tissue homeostasis. These processes could be a major contributing factor to aging and even cancer formation [93,95,96]. A robust body of evidence supports the occurrence of increased mtDNA damage and mitochondrial dysfunction in different cell types of HIV-infected individuals, including neuronal cells [97,98,99,100,101,102,103]. It is worth noting that mtDNA damage may be associated with HIV-related neurocognitive disorders [100,101].

The immune system in the context of HIV infection is severely weakened in individuals who progress to AIDS, usually in the absence of ART. However, even in individuals who receive adequate treatment and maintain an undetectable viral load, there are important and broad health consequences that cannot be overlooked [69,104]. In the context of immunosenescence, there is an acceleration of the processes that lead to the premature aging of the immune system, mainly because there is an increased activation of the response mechanisms due to the infection and associated features (e.g., co-infections, inflammation, and comorbidities). This increased activation results in greater cell replication and death associated with antiviral responses, which lead to the exhaustion of the immune system [105,106]. Furthermore, even the main pathway that leads to immunosenescence, which is thymus involution, is present at an early stage in HIV-infected individuals, as well as other damages in tissues that are important for the proper functioning of the immune system, such as bone marrow and liver tissue. Taken together, there is a prominent and premature aging of the immune system in HIV-infected individuals, with young people sometimes presenting clinical signs or immune features of individuals in their 40s [69]. This reverberates on multiple health aspects, including increased chromosome instability and associated risks.

## 4. Conclusions

The impact of HIV infection on chronic inflammation and chromosome instability is an emerging topic, especially because HIV infection is currently considered a chronic disease in many countries. In conclusion, both HIV and ART can cause chromosome instability, leading to MN formation and other chromosome aberrations (Table 1). Inflammation and immunosenescence are both the cause and consequence of chromosome instability. HIV infection also triggers inflammation by other mechanisms. Age-related diseases result, directly or indirectly, from (I) chromosome instability, (II) inflammation, and (III) HIV infection. These connections are summarized in Figure 2. Of note, these connections must be interpreted while taking into account classic environmental and lifestyle factors that can also trigger chromosomal instability, inflammation, and associated diseases (smoking habits, heavy metals, pollutants, and obesity, among many others) [8,107,108,109]. Dietary and lifestyle interventions, including reduced exposure to genotoxins and an antioxidant-rich diet, could be considered to mitigate the deleterious effects of HIV infection on DNA integrity.

Finally, the number of human studies that evaluate chromosome instability in the context of HIV infection is limited, which is noteworthy. Additionally, in general, the sample size in such studies is small, which represents another limitation. New studies concerning these topics can contribute to the clinical management of HIV-infected individuals undergoing ART, reducing chromosome instability and related health consequences.

## Figures and Tables

**Figure 1 viruses-15-00155-f001:**
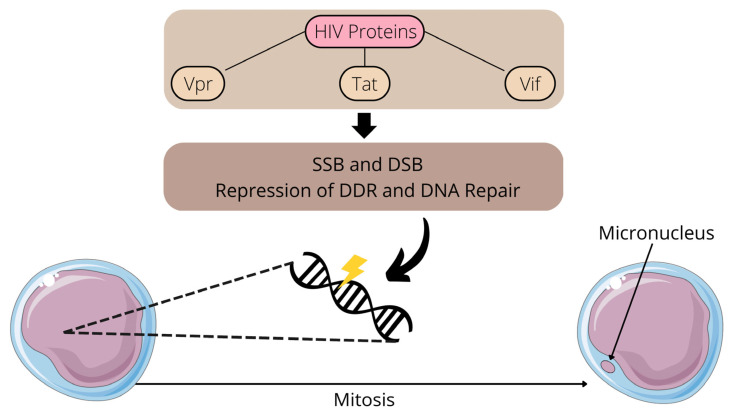
Effects of HIV proteins on DNA integrity and micronucleus formation. SSB: single-strand DNA break; DSB: double-strand DNA break; DDR: DNA damage response.

**Figure 2 viruses-15-00155-f002:**
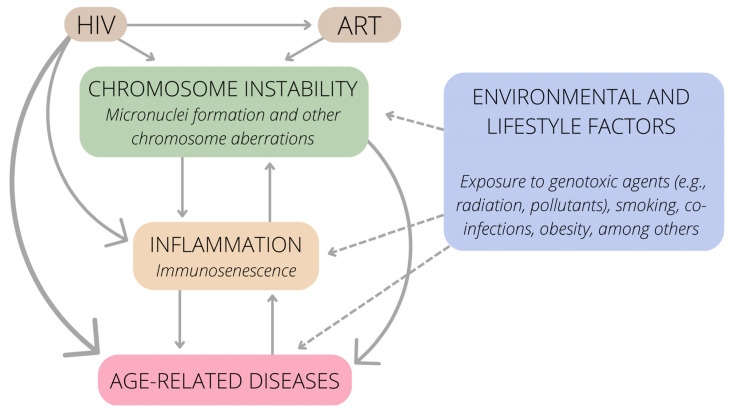
Connections between HIV infection, chromosome instability, inflammation, and age-related diseases.

**Table 1 viruses-15-00155-t001:** Summary of the impacts of HIV infection and treatment on chromosomal integrity based on studies with humans.

Country	Cell Type Investigated	Main Findings	References
Brazil	Exfoliated oral cells	Increased mean of single MN in cells of controls compared to those of HIV-infected individuals; non-significant increase in the occurrence of multiple micronuclei in cells of HIV group compared to controls	Lima et al. [32]
South Africa	Blood cells	MN frequencies were significantly higher in irradiated lymphocytes from HIV-infected individuals compared to controls	Baeyens et al. [33]; Herd et al. [34]; Herd et al. [35]
Italy	Blood cells	Increased MN frequency in the HIV-infected group (HCV co-infection and HIV-RNA load being risk factors for increased MN frequency); HIV-infected individuals with undetectable viremia showed reduced MN frequency compared to those with uncontrolled viremia	Zizza et al. [36]
USA	Reticulocytes	A 10-fold increase in micronucleated reticulocyte frequencies was observed in mothers and children pre-natal with zidovudine-based ART compared to controls	Witt et al. [42]
India	Exfoliated oral cells	HIV-infected individuals showed significantly increased in MN rate (~two-fold) compared to controls	Shah et al. [43]
Mexico	Exfoliated oral cells	Higher frequencies of binucleated cells and nuclear buds in both HIV-infected ART-naive individuals and HIV-infected individuals undergoing ART compared to HIV-negative controls; karyorrhexis, binucleated cells, and nuclear buds were found to be increased in a subgroup of HIV-infected individuals receiving RTIs as ART compared to controls	Gutiérrez-Sevilla et al. [44]
Mexico	Exfoliated oral cells	Increased frequency of MN and other nuclear abnormalities in HIV-infected individuals undergoing ART compared to controls	Lazarde-Ramos et al. [54]

MN: micronucleus. RTIs: reverse transcriptase inhibitors.

## Data Availability

Not applicable.

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
