# Peer review of "HIV Infection, Chromosome Instability, and Micronucleus Formation"

_viruses, 2023, doi:10.3390/v15010155_

Round 1

Reviewer 1 Report

This is a narrative review on chromosomal instability in HIV. 

My comments/questions are as follows:

1. Please provide citation after the sentence "Retroviruses damage DNA through multiple mechanisms... after inhibition of DNA replication."

2. The authors should include relevant articles on autophagy, inflammation, and cellular recycling.

3. Under the discussion on Immunosenescence, relevant studies on T cell exhaustion in HIV (PD-1, TIGIT, and TIM-3 surface markers) should me mentioned and their roles in immune dysregulation in chronic HIV.

4. The authors focused on nuclear DNA. However, the section on  mitochondrial DNA damage can further be expanded by providing relevant HIV-specific studies in this field.

5. Telomere length and epigenetics are key topics in HIV-related inflammation and aging. Please cite relevant literature on this.

6. Under 2.1 Please cite relevant literature that provide negative data/counter-acts the notion that antioxidants are beneficial in inflammation. For example, 'some selenoproteins may display harmful functions...' (Zhang, 2018-- Beneficial and paradoxical roles of selenium at nutritional levels of intake in healthspan and longevity'). In HIV, vitamin D has been shown to have a J-shaped curve effect on inflammatory markers (Gangcuangco, 2016 --High 25-hydroxyvitamin D is associated with unexpectedly high plasma inflammatory markers in HIV patients on antiretroviral therapy) and multiple studies have shown that supplementation with vitamin D has no effect on mortality. That said, the Abstract and Conclusions should be softened and the phrase 'anti-oxidant-rich diet should be encouraged' need to be revised as 'anti-oxidant-rich diet should be CONSIDERED.'

7. The study by Gutierrez-Sevilla showed that different ARVs may have various effects on chromosomal instability. Studies on newer antiretroviral agents on mitochondrial health and DNA damage should be cited (for example, dolutegravir (Ajaykumar, AIDS doi: 10.1097/QAD.0000000000003369). 

8. Figure 2 needs to be edited to reflect that age-related diseases and inflammation are bidirectional (for example, diabetes worsens inflammation and inflammation worsens diabetes in HIV). Other factors such as lifestyle, smoking, substance abuse, co-infections, obesity, etc. were also not reflected in this figure.

9. Suggest to include a table summarizing the various studies on chromosomal instability in HIV

Author Response

Comments from Reviewer 1

This is a narrative review on chromosomal instability in HIV.

My comments/questions are as follows:

  1. Please provide citation after the sentence "Retroviruses damage DNA through multiple mechanisms... after inhibition of DNA replication."

Response: Firstly, we would like to thank Reviewer 1 for the time dedicated to our manuscript and for the valuable comments. All changes in the manuscript are highlighted in yellow. Citation was added after the above-mentioned sentence, as requested.

  1. The authors should include relevant articles on autophagy, inflammation, and cellular recycling.

            Response: This is a good suggestion. Thank you. We have included a discussion on autophagy, cellular recycling, inflammation, and MN formation in the context of HIV infection. Please see the seventh paragraph of the Introduction section for details.

  1. Under the discussion on Immunosenescence, relevant studies on T cell exhaustion in HIV (PD-1, TIGIT, and TIM-3 surface markers) should me mentioned and their roles in immune dysregulation in chronic HIV.

Response: As requested, we have included (in the immunosenescence section, first paragraph) studies concerning T cell exhaustion and related markers in HIV infection. Thank you for this suggestion.

  1. The authors focused on nuclear DNA. However, the section on mitochondrial DNA damage can further be expanded by providing relevant HIV-specific studies in this field.

Response: Thank you for this suggestion. We added several studies on mitochondrial DNA damage and associated conditions in HIV-infected individuals. See these changes on page 8.

  1. Telomere length and epigenetics are key topics in HIV-related inflammation and aging. Please cite relevant literature on this.

Response: Thank you for highlighting this point. Following the Reviewer’s suggestion, we have included in the manuscript a discussion concerning telomere length and epigenetics in the context of HIV infection and aging (supported by several references). See these changes on page 8.

  1. Under 2.1 Please cite relevant literature that provide negative data/counter-acts the notion that antioxidants are beneficial in inflammation. For example, 'some selenoproteins may display harmful functions...' (Zhang, 2018-- Beneficial and paradoxical roles of selenium at nutritional levels of intake in healthspan and longevity'). In HIV, vitamin D has been shown to have a J-shaped curve effect on inflammatory markers (Gangcuangco, 2016 --High 25-hydroxyvitamin D is associated with unexpectedly high plasma inflammatory markers in HIV patients on antiretroviral therapy) and multiple studies have shown that supplementation with vitamin D has no effect on mortality. That said, the Abstract and Conclusions should be softened and the phrase 'anti-oxidant-rich diet should be encouraged' need to be revised as 'anti-oxidant-rich diet should be CONSIDERED.'

Response: Thank you for these important comments. In sub-section 2.1, we added a discussion concerning the deleterious effects associated with high doses of selenium, as well as the effects of vitamin D on inflammatory markers in HIV-infected individuals. We also mentioned that vitamin D supplementation in individuals with proper plasma levels of this micronutrient does not provide health benefits. We have cited in this discussion the suggested references and others. Finally, as requested, we revised the Abstract and Conclusions to soften the phrase mentioned by the Reviewer.

  1. The study by Gutierrez-Sevilla showed that different ARVs may have various effects on chromosomal instability. Studies on newer antiretroviral agents on mitochondrial health and DNA damage should be cited (for example, dolutegravir (Ajaykumar, AIDS doi: 10.1097/QAD.0000000000003369).

Response: Thank you for this suggestion. We have added to the manuscript a new paragraph discussing the effects of newer HIV drugs (e.g., Maraviroc, Dolutegravir) on DNA and mitochondrial health. We have cited the suggested manuscript (Ajaykumar et al.) and others. See these changes on pages 5-6.

  1. Figure 2 needs to be edited to reflect that age-related diseases and inflammation are bidirectional (for example, diabetes worsens inflammation and inflammation worsens diabetes in HIV). Other factors such as lifestyle, smoking, substance abuse, co-infections, obesity, etc. were also not reflected in this figure.

Response: Figure 2 has been modified following the Reviewer’s suggestions.

  1. Suggest to include a table summarizing the various studies on chromosomal instability in HIV

Response: As suggested, a table summarizing the impacts of HIV infection and treatment on chromosomal integrity (based on studies with humans) was added to the manuscript. Finally, thank you again for all comments and insights.

Reviewer 2 Report

Dear Authors,

great job on combining worldwide data on micronuclei formation during HIV infection.There are a great number of studies on inflammation development during HIV, however your review is more about genomic instability using micronucleus assay. Therefore, I would probably remove the word "inflammation" from the title and would add micronuclei formation or so. The title is very broad, and I think is a good idea to make it more focused on the topic you are describing. However, is your choice. Also, in updated review I would mostly use the latest publications from the past 10 years or so. Overall, this review is well written and combine a broad number of studies in one place. I recommend this review publishing in Viruses.

Author Response

Comments from Reviewer 2

Dear Authors,

great job on combining worldwide data on micronuclei formation during HIV infection. There are a great number of studies on inflammation development during HIV, however your review is more about genomic instability using micronucleus assay. Therefore, I would probably remove the word "inflammation" from the title and would add micronuclei formation or so. The title is very broad, and I think is a good idea to make it more focused on the topic you are describing. However, is your choice. Also, in updated review I would mostly use the latest publications from the past 10 years or so. Overall, this review is well written and combine a broad number of studies in one place. I recommend this review publishing in Viruses.

Response: First, we thank Reviewer 2 for the favorable feedback on our manuscript and suggestions. We changed the title to “HIV infection, chromosome instability and micronucleus formation”, as suggested. We also removed “an update” from the title to be consistent with the articles we cited in the manuscript (indeed not all cited publications from the past 10 years). All changes made to the text are highlighted in yellow.

Reviewer 3 Report

Ellwanger and colleagues review literature eon the effect of chronic HIV infection and combined anti-retroviral therapy (cART) on chromosomal instability and the appearance of micronuclei (MN). The topic seems of importance given that many people living with HIV (PWH) take cART, achieve viral suppression and survive long-term and come of age. Yet, even virally well controlled PWH appear to have a higher risk of developing diseases associated with chromosomal instability, in particular a variety of cancers, compared to the uninfected part of the population. Oxidative stress and aging-related inflammation are discussed as prime candidates driving chromosomal instability and some studies indicate that diet and lifestyle can influence the development and extent of MN formation. The review is overall well written and referenced. The authors also acknowledge limitations of the studies described in the current literature, such as low numbers of human subjects. Altogether, a concise and informative review.

Author Response

Comments from Reviewer 3

Ellwanger and colleagues review literature eon the effect of chronic HIV infection and combined anti-retroviral therapy (cART) on chromosomal instability and the appearance of micronuclei (MN). The topic seems of importance given that many people living with HIV (PWH) take cART, achieve viral suppression and survive long-term and come of age. Yet, even virally well controlled PWH appear to have a higher risk of developing diseases associated with chromosomal instability, in particular a variety of cancers, compared to the uninfected part of the population. Oxidative stress and aging-related inflammation are discussed as prime candidates driving chromosomal instability and some studies indicate that diet and lifestyle can influence the development and extent of MN formation. The review is overall well written and referenced. The authors also acknowledge limitations of the studies described in the current literature, such as low numbers of human subjects. Altogether, a concise and informative review.

Response: We thank Reviewer 3 for the time devoted to reviewing our manuscript and for the kind and consistent comments. We are glad about this good feedback.

Round 2

Reviewer 1 Report

The authors have adequately addressed all my comments and inquiries. They have included relevant literature to provide a comprehensive discussion on HIV infection and chromosome instability. I recommend to accept the paper for publication.